# Multistage Models for Flood Control by Gated Spillway: Application to Karkheh Dam

Farhad Salehi [1], Mohsen Najarchi [1,*], Mohammad Mahdi Najafizadeh [2] and Mohammad Mirhoseini Hezaveh [1]

[1] Department of Civil Engineering, Faculty of Engineering, Islamic Azad University of Arak Branch, Arak 38135-567, Iran; farhadsalehi2005@gmail.com (F.S.); m-mirhoseini@iau-arak.ac.ir (M.M.H.)
[2] Department of Mechanical Engineering, Faculty of Engineering, Islamic Azad University of Arak Branch, Arak 38135-567, Iran; mohammadnajafizadeh@yahoo.com
* Correspondence: m-najarchi@iau-arak.ac.ir

**Abstract:** The paper demonstrates a simulation optimization framework for enhancing the real-time flood control with gated spillways at places where no flood forecasting data are available. A multiobjective modeling scheme is presented for the flood management in a gated spillway in which the operator may specify the priorities on floods based on their different return periods. Two different operation strategies were devised. Both operating strategies employ ten-stage policies, which rely on the reservoir water level as the input data. The second strategy benefits from both the observed reservoir water level and the flood peak. The optimal values of the models' parameters were obtained using a genetic algorithm. This is a novel approach because none of its policies needs flood forecasting data, thus, making them adaptable to any flood with any return period. To evaluate the performances of the proposed models, the flood control through a gated spillway of the Karkheh reservoir was considered, where flood hydrographs with different return periods were routed through the reservoir.

**Keywords:** flood hydrograph; gated spillway; multistage; Karkheh reservoir

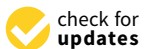



## 1. Introduction

Reservoir flood control and management can be categorized as real-time operation problems, and significantly differ from long-term planning and operation. The real-time reservoir operation for flood management involves the operation of a reservoir system in relatively short time intervals by taking decisions on reservoir releases based on the information available [1]. Ideally, an optimum gate opening and reservoir operation strategy should minimize downstream damage, i.e., maximize the volume of water stored at the end of the flood event; and ensure the safety of the dam within reasonable limits.

Real-time reservoir operation models for flood control may or may not involve real-time data external to the reservoir. Most of the existing models benefit from flood forecasting and data external to the reservoir in one way or another. In this category, [1] developed and applied a model for the real-time optimal flood operation of river reservoir systems by using a combination of a nonlinear programming model and a flood-routing simulation model within an optimal control framework. Braga and Barbosa [2] developed a linear programming (LP) deterministic optimization model to solve the problem of flood control in real time for a multireservoir operation system benefiting the Muskingum method of channel routing. Using the balanced water level index in a real optimization operation procedure, [3] integrated a hydrologic forecasting model with a reservoir operation scheme to determine the reservoir releases during a flood at each time step.

The work in [4] suggested a weighted preemptive goal programming method to coordinate the reservoir operation during the flood. Combining the model with a multiple-inflow forecasting model enabled the model to be used for real-time reservoir operation. The authors in [5] developed a real-time operation model for determining the optimal reservoir release during a typhoon, in which two forecast modules were integrated with a

reservoir optimization model to reduce the peak flow in selected downstream checkpoints while optimizing the reservoir storage when the flood subsides. In addition, the authors in [6] proposed a multicriteria analysis of the Irkutsk reservoir operating methods under different hydrological conditions considering dispatch schedules and optimization methods. Moreover, Bolgov, M.V. et al. [7] proposed an approach to search for compromise decisions in water resources management in the interests of various water users. To effectively address the importance of dam safety during rare and devastating floods, [8] presented a control method based on a fuzzy logic controller and a tabu search (TS) algorithm for the operation of a dam having a gated spillway during the probable maximum flood (PMF) occurrence. Since most of the existing flood control management systems for reservoirs are established for special purposes, Cheng and Chau [9] presented a national flood control management system to overcome the various problems associated with receiving real-time information in a short time due to the lack of data sharing and communication with the authorized agencies. They recommended that interactive interfaces should be designed to generate effective alternatives for flood control operation. However, it should be noted that all these approaches either demand flood forecasting modules or assume a given flood event with a predefined hydrograph in one way or another. In [10], a simulation optimization method in a real-time flood control utilization of river reservoir systems was developed. The work presented the development and testing of a methodology for determining reservoir release schedules before, during, and after a massive flood occurrence in real time.

The operation of the dam having a gated spillway was simulated by employing the volumetric evaluation method (VEM), fully described in [11]. Recent research on new reservoir operation models by Sun et al. [12] used the tree-based stochastic optimization method to propose a risk analysis model for reservoir real-time optimal operation. Neither of these models' interfaces optimization and simulation to solve an optimal control problem for specifying the real-time gate control operations of river reservoir systems. The work in [13] presented a predictive-model-based control scheme for the real-time operation of a multireservoir system in the Sittaung river basin in Myanmar. The control objectives were to minimize the storage deviations in the reservoirs, to minimize flood risks at a downstream location as a vulnerable place, and to maximize hydropower generation by using penalty coefficients to water levels and flows of the system in deviation from objectives.

Cuevas-Velasquez et al. [14] presented a real-time flood operation model for dams having gated spillways that brought together the benefits of an optimization model based on mixed integer linear programming (MILP) and a case-based learning method using Bayesian networks (BNs). In reservoir operation during flood events, the human operators regulate the spillway gates opening according to the prevailing flood conditions. With uncertainties involved in the prediction of inflow flood hydrograph, it is highly difficult to construct and implement a precise inflow–outflow rule for reservoir operation under flooding condition. Therefore, the implementation of simulation-based and/or heuristic conventional operation rules may not efficiently use the existing flood control volumes to reduce the flood peak downstream or may result in sharp variations in outflow with respect to time. In general, it is recommended to make the time variation of the released water as smooth as possible while keeping the water surface elevation within a prescribed range. These requirements are the fundamental goals of any reservoir control approach where reservoir operation is a complex, nonlinear, and nonstationary control problem [8]. Computational fluid dynamics (CFD) is a powerful tool that makes it possible to use numerical methods and turbulence models to simulate complex three-dimensional (3D) flow fields. CFD has gradually been accepted as an independent and reliable data source along with field- and laboratory-based measurements for many studies in hydraulics and fluid mechanics phenomena [15,16].

Due to the importance of dam safety, the operation of a gate spillway remains an obvious challenge in reservoir management and operation during flood event. In [17], a flood routing method for gated spillways was presented, which involved the implementation of

a six-stage operation policy for the routing of flood hydrographs ranging from very small magnitudes up to the probable maximum incoming flood to the reservoir. Later, ref. [18] described a set of operation rules with ten stages for controlling the spillway gate opening to discharge a constant amount of water according to the reservoir level.

Flood control by gated spillways is a continuous and instant decision-making process based on relevant operating rules, policy, and physical laws. The decision variables are the release values (or gate openings) for the flood operation period of the reservoir. Hence, the determination of water levels and releases from the reservoir that satisfies various operating needs and restrictions such as water balance equations, upper and lower bounds on release and storage, as well as general standards of operating procedure, may be realized as the main purpose of any flood operation program. This approach formulates the reservoir flood operation as an optimization problem with perfect knowledge of the hydrograph and using a genetic algorithm (GA). However, ref. [19] introduced an early warning model for real-time reservoir operation during typhoons. The work defined a flood alert index to consider the risk; it was used to augment the reservoir flood operation. A GA-based algorithm was derived to specify the appropriate releases in response to the nature of the flood inflows and reservoir water level. Their model benefitted from a flood watch, flood release, and decision analysis. Although not much related to any data external to the reservoir itself, the flood watch element of this model was designed to monitor the current flood situation externally to the reservoir. Haktanir et al. [20] proposed a procedure to identify sets of operational rules for gated spillways for the optimal flood routing management of artificial reservoirs. In their work, the flood retention storage of a dam having a gated flood spillway was divided into 15 substorages where the critical levels were the surface elevations. Moreover, ref. [21] proposed a real-time control tool for modeling the real-time control and decision support in water resources systems. It combined various control paradigms ranging from simple feedback control strategies with triggers, operating rules, and controllers to advanced optimization methodologies such as model predictive control (MPC). Furthermore, a multiobjective optimal operation of gated spillways was proposed in [22] where, in each stage, the opening of the gates was proportional to the water level of the reservoir. A GA was implemented for determining the optimal opening of the gates to minimize downstream damages. More recently, ref. [23] developed an inflow flood forecasting based on a distributed hydrological model for the Baipenzhu Reservoir in the Guangdong Province of China.

This article presents an optimized and practical operating rule for the gated spillways in a real-time-based operation for floods ranging from very small magnitude to the PMF. This is a novel approach for designing gated spillways operation policies, which can be used when flood forecasting is not available. Thus, it is adaptable to any flood of any magnitude without any information on the incoming flood. In the absence of reliable flood prediction, the safe and efficient operation of a gated spillway is a challenge. Assuming that the actual magnitude of a flood cannot be predicted beforehand, the objective of this study is to design a set of spillway operation rules that will route floods of all magnitudes through the reservoir safely and efficiently while minimizing the human errors caused by decision making under stress during flood operations. In addition, this article introduces a modeling framework for enhancing real-time flood control with gated spillways where no flood forecasting data are available. The proposed approach improves the simple ten-stage strategy (TSS) that was proposed and tested by Haktanir and Kisi [18] in a simulation optimization framework. The TSS is used as the reference operation policy to compare the policies proposed in this research. The scheme presented in this paper considers the flood management in a gated spillway as a multiobjective optimization problem in which the operator's objective is to minimize the flood damage for all floods with a different probability of occurrence. In this paper, the optimization problem is formulated under the framework of a genetic algorithm. In the absence of reliable economic data on flood damages, the proposed approach relies on the priorities assigned by the operator or the decision makers to floods with different return periods. Two different operation policies

were tested, and the operator's priorities were considered as weights to different objectives. The first operation policy (OP1) employs a ten-stage policy that only relies on the reservoir water level and spillway capacity/behavior.

The second operation policy (OP2) considers the observed reservoir water level and visual information on the flood peak at the connection of the river to the reservoir to define the release policy. In addition, we assume that a system for transferring measured data to the control site is available. Thus, the information about the flood peak would be available. The models intend to provide the operator with the optimal values for the control levels and their associated gate openings. Both approaches can be used for real-time operation systems and may be easily practiced by operators with a general knowledge of the operation of the system.

## 2. Methodology

### 2.1. Rule Curve Flood Management in Gated Spillways

In comparison with ungated spillways, gated spillways provide more operation flexibility by incorporating several strategies in passing extreme floods. The maximum desirable rate of downstream release can be used to develop the operation policy for the spillway.

Rule curves are decision tools in the form of equations and/or graphs relating the spillway gate openings to the reservoir state parameters. Furthermore, rule curves are independent of data external to the reservoir itself. Generally speaking, one may either adopt a closing or an opening strategy. If the gate openings are kept too small at the rising limb of a serious flood and a "closing" strategy is adopted [24], the peaks released later may be greater than they would under a better policy or even exceed the natural flood peak. On the other hand, if an "opening" strategy is adopted, significant flows may be released in the early stages of flood or even in advance. In this case, unnecessarily large gates opening at the onset of the flood may result in outflow peaks greater than they would have if a tighter policy with full use of the flood retention storage and the spillway characteristics in proportion to the real strength of the incoming wave were adopted. Being safer for dam overtopping, the opening strategies are often adopted for high dams. Rule curves are usually developed by employing a simulation model in which critical levels and gate openings are optimized by using appropriate optimization algorithms.

Engineers often prefer to rely on simple, well-written, and documented action lines rather than complex computer algorithms. Therefore, developing simple operating rules which the operator can follow in deciding on how to control the gates opening at different stages of flooding is important. Typically, the system is managed to minimize the flood peak at the protection site and to avoid exceeding the channel capacity. Thus, a function of the recent water surface elevation in the lake and inflow in the upstream gagging station can be used to determine the gate openings of the operable spillway. It would be practical if a fixed set of optimum operation rules could be developed based on the variation of the lake level.

As a good example of an easily understood and well-documented simple operating rule, one may refer to [18]. Improving on their previous models, the authors presented a ten-stage operation policy for the routing of flood hydrographs with return periods from 10% PMF, 20% PMF up to the PMF for any dam having a gated spillway. The generalized gate opening rules are determined depending on the recent pool level. Regardless of the size and timing of any incoming floods, the fixed rules of the ten-stage operation policy may provide semioptimal routing for all floods, which are classified into ten different groups based on the PMF's volume. The upper limit of each group is identified by a predefined percentage of the PMF. Thus, 10% PMF, 20% PMF, 30% PMF, and so on, are the upper limits for the ten groups. Small floods, having return periods less than or equal to 10% of the PMF, are effectively routed within the first stage. The choice of the ten stages is claimed to be reasonable, as more stages would make the gate operations complicated during the fairly short period of a particular flood, and fewer stages would reduce the operation

accuracy. Associated with each stage (critical level), a reservoir volume may be defined as follows [18]:

$$S_{cr(j)} = S_{cr(j-1)} + \left(S_{ult} - S_{cr(j-1)}\right)/(10 - (j-1)) \tag{1}$$

where $S_{cr(j)}$ is the reservoir volume (m$^3$) for critical level $j$, $S_{cr(j-1)}$ is the reservoir volume (m$^3$) for critical level $j-1$, and $S_{ult}$ is the maximum reservoir volume (m$^3$) for the last critical level or the tenth critical level ($H_{cr}^{10}$) (m).

These ten critical levels (stages) have crucial roles to determine the rate of gate opening at each stage and the flood retention in the reservoir. The proposed operation rule assigns a smaller and more gradual gate opening for small floods with higher frequencies, in the initial stages of the flooding rather the upper stages of flooding. Therefore, these floods may be efficiently stored in the flood storage volume of the reservoir.

*2.2. Proposed Optimization Model for Gate Openings*

This part intends to describe the structure of the optimization model that minimizes the flood peak with different return periods. Realizing the stochastic nature of the floods, one may minimize the expected damage or the mathematical expectation of the losses. As outlined by [25], while ensuring the dam safety, one may consider different flood operation goals such as postponing the arrival time of the flood peak, minimizing the release peak, and keeping the final water level close to the predefined target storage. While the final water level should reach the target level, the main index in evaluating the effect of the flood operation is the extent of the reduction of the flood peak [26].

Generally, one may consider the flood management by gate opening as a multiobjective optimization problem in which the operator wishes to minimize the flood damage for all floods with different probability of occurrence. In a mathematical statement and for a discrete version of the modeling approach, the multiple-objective gate opening for flood management may be presented as follows:

$$\text{Min} f_i(Q_i(peak)); \; i = 1, 2, \ldots n \tag{2}$$

where $Q_i(peak)$ refers to the peak outflow (m$^3$/s) from the reservoir for the discrete probability level of $i$ and $f_i$ refers to the function that approximates the damage caused by the flood with a peak of $Q_i(peak)$. In an ideal condition, the operator may wish to release floods from the reservoir to minimize the damage for all possible floods with a generalized gate opening rule. In other words, the operator may wish to maximize the percent of peak reduction for all floods with different return periods. This problem may be solved by transferring the multiple objectives into a single weighted objective function, where different weights may be assigned to different objectives to reflect their importance in the damage management process.

In a general statement, one may assume that the expected flood damage ($ED$ (m$^3$/s)) may be approximated as a function of flood peak and the occurrence probabilities as:

$$ED = \sum P_i \times f(Q_i) \tag{3}$$

where $f(Q)$ is the flood damage for a peak of $Q$ (m$^3$/s) and $P_i$ is the probability of occurrence. Therefore, the weighted objective function may be defined as:

$$Min(ED) = w_1 Q_1'(peak) + w_2 Q_2'(peak) + \cdots + w_k Q_k'(peak) \tag{4}$$

$$\sum w_i = 1$$

where $Q_k'(peak)$ is the peak outflow (m$^3$/s) corresponding to the inflow flood with a discrete probability level $k$, and $w_i$ the weights assigned to the objective, which should represent its importance in flood damage reduction management.

The main inequality constraint which needs be handled with a special methodology is the one that bounds the reservoir water level after routing the floods of different return

periods with the generated trial solutions. The well-known penalized objective function method was employed for constraint handling to eliminate the chance of reservoir overtopping. Therefore, the objective function defined by Equation (4) was replaced by the penalized objective function as:

$$Min(ED) = w_1 Q'_1(peak) + w_2 Q'_1(peak) + \cdots + w_k Q'_k(peak) + P * \sum (overtopping\ height)_k \tag{5}$$

$$Overtopping\ height = 0,\ if\ RWL < IMWL$$
$$Overtopping\ height = RWL\text{-}IMWL,\ if\ RWL > or = IMWL$$

where $P$ is a balancing coefficient to be tuned for pushing the solutions toward the feasible ones. In this study, it was selected as 10,000. RWL and IMWL are the reservoir water level (m), and maximum reservoir flood level (m), respectively. Without loss of generality, this study employed three different approaches for assigning appropriate values to outflow floods from the reservoir with return periods of 10, 20, 50, 100, 500, 1000, and 10,000 years. The values for the weight coefficients were based on floods probabilities of occurrence as presented in Table 1. The weight coefficients were derived using the following exponential relation for different values of $n$ as presented in Table 1.

$$\frac{P_i^{1/n}}{\sum P_i^{1/n}} \tag{6}$$

where $n$ is a parameter selected by the operator to reflect the relative importance of the flood control strategy for low and high return period floods or flood damage cost function. It should be noted that smaller values of $n$ assign higher relative priorities for floods with low return periods, whereas the larger values for $n$ reduce the importance of low return period floods by assigning smaller weights (Table 1).

**Table 1.** Proposed weights based on floods probabilities of occurrence in three different states.

| Return Periods | Weights ($n = 1$) | Weights ($n = 2$) | Weights ($n = 3$) |
|---|---|---|---|
| 10 | 0.9 | 0.6004 | 0.4369 |
| 20 | 0.05 | 0.1415 | 0.1667 |
| 50 | 0.03 | 0.1096 | 0.1406 |
| 100 | 0.01 | 0.0632 | 0.0975 |
| 500 | 0.009 | 0.0600 | 0.0941 |
| 1000 | 0.0009 | 0.0189 | 0.0436 |
| 10,000 | 0.00009 | 0.0060 | 0.0202 |

The objective function defined by Equation (5) is subject to different constraints and may be minimized with the optimal selection of the decision variables. This study considered the critical levels (Hcr1, Hcr2, ... , Hcr10) and gate opening at each critical level (D1, D2, ... , D10) as decision variables. Without loss of generality, this study assumed ten control (or critical) levels for the generalized gate operation rule. Appropriate values of the control levels and associated percent of gate openings minimize the weighted objective function defined by Equation (5). Equation (5) intends to minimize a measure of expected flood damages resulting from the generalized gate operation rule. In this case, the operator provides ten critical control levels and corresponding percentages of gate opening for each level, which may facilitate the operation. The proposed optimization model is subject to various technical and physical constraints mainly defined with the flood routing equations in the reservoir for gated spillways. The first ten constraints define the relationships between the control (critical) levels and the outflow resulting from the planned gate operation schemes:

$$Q_{out} = 0,\ if\ H_b \langle RWL \le H_{cr}^1 \tag{7}$$

$$Q_{out} = f(D_2, RWL), \ if \ H_{cr}^1 \langle RWL \leq H_{cr}^2 \tag{8}$$

$$Q_{out} = f(D_3, RWL), \ if \ H_{cr}^2 \langle RWL \leq H_{cr}^3 \tag{9}$$

$$Q_{out} = f(D_4, RWL), \ if \ H_{cr}^3 \langle RWL \leq H_{cr}^4 \tag{10}$$

$$Q_{out} = f(D_5, RWL), \ if \ H_{cr}^4 \langle RWL \leq H_{cr}^5 \tag{11}$$

$$Q_{out} = f(D_6, RWL), \ if \ H_{cr}^5 \langle RWL \leq H_{cr}^6 \tag{12}$$

$$Q_{out} = f(D_7, RWL), \ if \ H_{cr}^6 \langle RWL \leq H_{cr}^7 \tag{13}$$

$$Q_{out} = f(D_8, RWL), \ if \ H_{cr}^7 \langle RWL \leq H_{cr}^8 \tag{14}$$

$$Q_{out} = f(D_9, RWL), \ if \ H_{cr}^8 \langle RWL \leq H_{cr}^9 \tag{15}$$

$$Q_{out} = f(D_{10}, RWL), \ if \ H_{cr}^9 \langle RWL \leq H_{cr}^{10} \tag{16}$$

$$Q_{out} = f(D_{fullyopen}.RWL), \ if \ RWL \geq IMWL \tag{17}$$

where $H_b$(m), $H_{cr}^{1-10}$(m), $D_{1-10}$(m), $Q_{out}$(m$^3$/s), $D_{fullyopen}$(m), $RWL$(m), $IMWL$(m) are the threshold level, maximum water surface elevation for critical levels 1–10, gate opening, discharge, fully opened gate opening, reservoir water level, and maximum water level for considering dam safety, respectively.

$$\frac{dS}{dT} = I - Q \tag{18}$$

where $\frac{dS}{dT}$, $I$, and $Q$ are the rate of change of storage, inflow, and outflow, respectively. The discharge from the gated spillway may be estimated as:

$$Q = 1.63 * C * L * \left[ H^{3/2} - (H - D)^{3/2} \right] \tag{19}$$

where $Q$(m$^3$/s), $C$, $L$(m), $H$(m), and $D$(m) are the discharge, spillway discharge coefficient, length of spillway, head on the spillway crest, and gate opening, respectively.

Having formulated the objective function, one should merge the appropriate optimization method with the simulator model to determine the optimal critical levels and spillway gate openings. Since differentiating the objective function is rather complicated, we disregarded the classic gradient-based optimization algorithms such as nonlinear programming. On the other hand, indirect search optimization techniques usually bear small convergence rates, and they require executing considerable evaluations of the objective function, which may cause an ever-increasing calculation cost. Furthermore, the efficiency of these algorithms greatly relies on the nature of each optimization issue, while they are prone to be trapped in the local optimization spaces. Thus, an elitist genetic algorithm was used in this study. The selection and pairing operators were the roulette wheel and two-point crossover, respectively. A penalty technique was employed for constraint handling to prevent overtopping. The conditional optimization was changed to an unconditional issue by the proposed method.

### 2.3. Genetic Algorithm

A genetic algorithm is a statistical search method based on population genetics. A set of solution called population is required to start the algorithm and the solution is represented by a chromosome. The population size is preserved during each generation. At each generation, the fitness of each chromosome is evaluated, and then chromosomes for the next generation are probabilistically selected according to their fitness values. Some of the

selected chromosomes randomly mate and produce offspring. When producing offspring, crossover and mutation randomly occur. Since chromosomes with high fitness values have a high probability of being selected, chromosomes of the new generation may have a higher average fitness value than those of the old generation. The process of evolution is repeated until the convergence criteria are met. Taking advantage of this algorithm and considering a population with a size of 80, the optimal values for the unknown parameters were obtained.

Figure 1 illustrates a schematic interaction between different elements of the developed simulation and optimization model. Employing results of the optimization model defined by Equations (4)–(19), two different operation strategies were structured. The first operation policy (OP1) employs Equation (4) ten-stage policy, which relies on the reservoir water level and spillway capacity where its control levels and their associated gate openings are extracted from the optimization model. In the second operation policy (OP2), however, both observed reservoir water level and flood peak in the upstream gaging station define the release policy. In this strategy, as for OP1, the rates of gate opening and the critical control levels are determined by the proposed simulation optimization model; however, the operating strategy is somehow different. In addition to the optimization model results, the OP2 strategy benefits from a tiny and available information on the timing of the peak of the incoming flood. This strategy receives the information from the gaging station on whether the flood's peak has passed the gaging station. This helps the operator to enhance their operation and discharge the flood with a smaller gate opening and downstream flooding. In this case (OP2), during the application of a command control discharge, when a new critical flood level is observed, two different strategies may be employed. The operation is continued as planned if the peak of flood has not yet passed the gaging station. On the other hand, if the peak has already passed the gaging station and reached the reservoir, the gate opening is retarded until the water level reaches the next critical stage. This means that the gate opening is kept as for the previous time step and is not increased even if the water level in the reservoir is rising. This condition is continued until the water level approaches the next critical level. Then, the gate opening is increased to the next level provided that the water level meets the next associated critical stage. This scheme is followed until the outflow from the reservoir exceeds the inflow; that is, when the water level in the reservoir falls down. For the sake of safety, the gates are completely opened if the water level approaches the last critical stage.

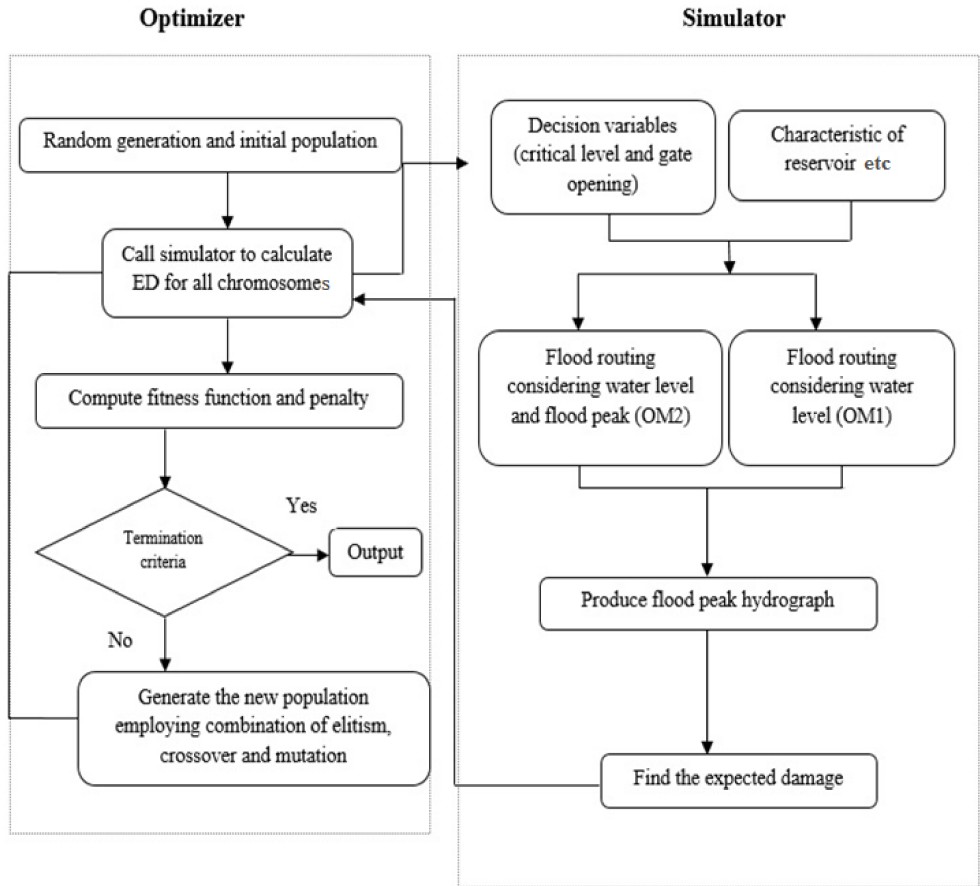

**Figure 1.** Flowchart of a genetic-algorithm-based simulation-optimization model of gated spillway.

## 3. Results

In this section, a case study is introduced to validate the efficiency and applicability of the proposed models. The proposed models were implemented to upgrade the spillway operation and gate opening rules for the Karkheh dam in Iran. The Karkheh dam in the south-west of Iran, which is the only dam in the river basin, is a multi-purpose embankment dam with a 127 m height and a total storage volume of approximately $59 \times 10^8$ m$^3$. Its main objective is to irrigate $32 \times 10^4$ hectares of irrigable land. Hydropower production and flood control have been referred to as its second and third objectives. The spillway is equipped with six similar $18 \times 15$ m gates installed on a chute spillway. The spillway crest is located at 209 m.a.s.l and the maximum flood level of the reservoir has been fixed at 234 m.a.s.l. The spillway is designed to discharge a 10,000-year flood with a 1.2 m freeboard. Estimated flood hydrographs with different return periods are provided in Figure 2.

This application intends to test the performance of the proposed operation schemes defined as OP1 and OP2 and compare the results with those of the simple ten-stage simulation model of [18].

In this study, the population size was set to 80. Moreover, the number of elites and mutation probabilities were set to 1 and 0.015, respectively. The results of implementing the simple ten-stage simulation model [18] in addition to the first- and second-level optimization model (OP1 and OP2) are summarized in Table 2. This table also demonstrates the ten critical levels along with the associated gate openings in optimal conditions.

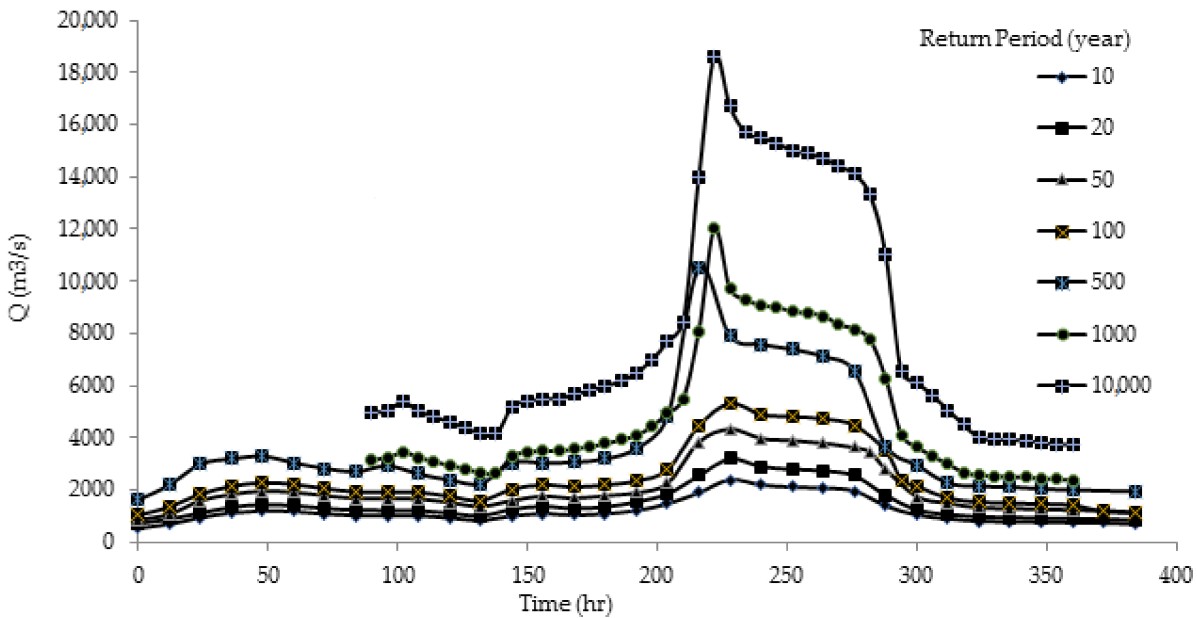

**Figure 2.** Flood hydrographs of the Karkheh reservoir (adapted from consulting engineering company of Mahab Ghods).

**Table 2.** Critical levels and gate opening—Karkheh dam ($n$ = 3).

| Critical Levels (m) | Haktanir and Kisi [18] | OP1-GA | OP2-GA | Gate Opening (m) | Haktanir and Kisi [18] | OP1-GA | OP2-GA |
|---|---|---|---|---|---|---|---|
| Hb | 209 | 209 | 209 | D1 | 0 | 0 | 0 |
| Hcr1 | 212.11 | 211.25 | 210.25 | D2 | 1.87 | 2 | 0.75 |
| Hcr2 | 214.97 | 214.75 | 214.75 | D3 | 2.8 | 2 | 2.05 |
| Hcr3 | 217.88 | 217.25 | 217.25 | D4 | 3.98 | 3 | 3 |
| Hcr4 | 220.64 | 219.75 | 218.75 | D5 | 3.98 | 4.85 | 4.25 |
| Hcr5 | 222.9 | 220.25 | 220.25 | D6 | 5.9 | 6.25 | 6.25 |
| Hcr6 | 225.31 | 224.75 | 223.75 | D7 | 6.3 | 9.25 | 9.25 |
| Hcr7 | 227.64 | 225.25 | 227.25 | D8 | 6.3 | 10.25 | 9.85 |
| Hcr8 | 229.6 | 228.75 | 229.75 | D9 | 6.3 | 10.75 | 11.05 |
| Hcr9 | 231.73 | 231.25 | 230.25 | D10 | 8.02 | 12.55 | 13.75 |
| Hcr10 | 234 | 233.75 | 232.75 | - | - | - | - |

The results obtained from operating the Karkheh dam spillway at floods with 10-to-10,000-year return periods are presented in Table 3, adapting instructions from the simple simulation model. As can be observed, it is possible to discharge a flood with a 10,000-year return period with a 8.02 m gate opening and a maximum upcoming of the water surface elevation of 24.12 m. Thereby, the flood peak decreases by 47.33% and drops from 18,481 m³/s to 9734 m³/s. The percentage of the mitigated outflow flood vary from 31.29% to 50.92% for different return periods, while the total value of the cumulative weighted value of the objective function is approximated as 2726 m³/s for return periods of 10 to 10,000 years.

However, by using the instructions obtained from the first-level optimization model for $n$ = 3, the total cumulative weighted objective function does not exceed 2488 m³/s for floods with 10 to 10,000-year return period. In this case, flood peak reductions vary from 26.79% to 55.95%, which is indicative of a completely more efficient behavior compared to the simple ten-stage simulation model. Meanwhile, by using the instructions obtained from the second-level optimization model, the total values of the cumulative weighted objective function do not exceed 1802 m³/s for floods with 10-to-10,000-year return periods and the

reduction in the flood peak approaches 50% and 74.5%, respectively. This also indicates a more efficient behavior for the commands of the second-level optimization model.

**Table 3.** Results for ten-stage simulation model of Haktanir and Kisi [18].

| Return Period | Time (hr) | Peak Inflow (m³/s) | Peak Outflow (m³/s) | Max Water Level (m) | Max Gate Opening (m) | Flood Reduction (%) | Objective Function Value (m³/s) |
|---|---|---|---|---|---|---|---|
| 10 | 283.5 | 2387 | 1640 | 6.1 | 2.8 | 31.29 | 716.43 |
| 20 | 286.5 | 3181 | 1847 | 7.35 | 2.8 | 41.94 | 308.02 |
| 50 | 286.5 | 4317 | 2944 | 9.48 | 3.98 | 31.80 | 414.05 |
| 100 | 289.5 | 5293 | 3236 | 11.02 | 3.98 | 38.86 | 315.52 |
| 500 | 280.5 | 10,337 | 5524 | 14.94 | 5.9 | 46.56 | 520.10 |
| 1000 | 289.5 | 11,856 | 5819 | 16.24 | 5.9 | 50.92 | 254.28 |
| 10,000 | 289.5 | 18,481 | 9734 | 24.12 | 8.02 | 47.33 | 197.43 |
| Sum | - | - | - | - | - | - | 2726 |

The percentage of peak reduction for any flood with a different return period is highly dependent on the weight assigned to the different objectives (Table 1). For OP1 and values of weights corresponding to $n = 3$, the peak reduction for floods with return periods up to 500 years exceeds those of the simple ten-stage simulation model. For OP2, however, the peak reduction far exceeds those of the simple ten-stage simulation model for all return periods.

Therefore, by comparing the results obtained from the ten-stage simple simulation model [18] and the ten-stage models optimized by genetic algorithm (OP1 and OP2), it was observed that the later imposed more mitigation degree on the inflow flood to the reservoir. Thus, the OP1 and OP2 strategies of the gated spillways provide much more flexibility in operational strategy under different flooding conditions. It is worth emphasizing that the operator will simply decide on the gate openings and discharge of flood with any return period based on the optimal critical levels and values of the proposed gate openings.

## 4. Discussion

Figure 3 depicts the peaks of the outflow flood with various return periods for 100 repetitions and a population of 80 according to Tables 3–9. It can be clearly seen that the answers obtained by applying the genetic algorithm (OP1 and OP2), especially those that have flood hydrographs with 10-, 20-, and 50-year return periods, have outperformed the model of Haktanir and Kisi [18].

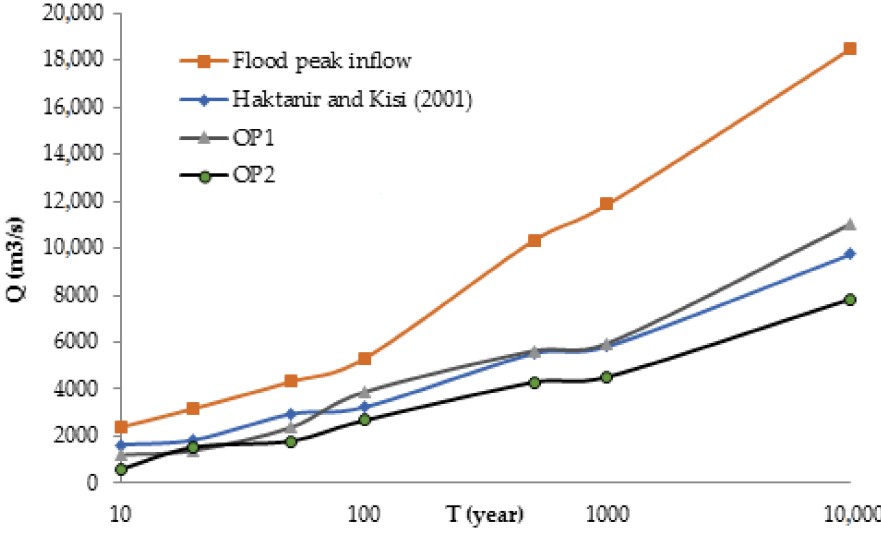

**Figure 3.** Comparison of flood peak with different return periods for different methods ($n = 3$).

**Table 4.** Optimized results from simulated optimization model OP1 (genes, 100; population, 80; *n* = 1).

| Return Period | Time (hr) | Peak Inflow (m$^3$/s) | Peak Outflow (m$^3$/s) | Max Water Level (m) | Max Gate Opening (m) | Flood Reduction (%) | Objective Function Value (m$^3$/s) |
|---|---|---|---|---|---|---|---|
| 10 | 288 | 2387 | 1433 | 5.81 | 2.48 | 39.99 | 1289.54 |
| 20 | 291 | 3181 | 1649 | 7.28 | 2.48 | 48.15 | 82.47 |
| 50 | 288 | 4317 | 2786 | 8.94 | 3.9 | 35.47 | 83.58 |
| 100 | 277.5 | 5293 | 4304 | 9.59 | 6.33 | 18.70 | 43.04 |
| 500 | 277.5 | 10,337 | 6246 | 13.16 | 7.58 | 39.58 | 56.21 |
| 1000 | 286.5 | 11,856 | 6670 | 14.44 | 7.58 | 43.74 | 6.00 |
| 10,000 | 285.75 | 18,481 | 11,843 | 20.53 | 11.42 | 35.92 | 1.07 |
| Sum | - | - | - | - | - | - | 1562 |

**Table 5.** Optimized results from simulated optimization model OP2 (genes, 100; population, 80; *n* = 1).

| Return Period | Time (hr) | Peak Inflow (m$^3$/s) | Peak Outflow (m$^3$/s) | Max Water Level (m) | Max Gate Opening (m) | Flood Reduction (%) | Objective Function Value (m$^3$/s) |
|---|---|---|---|---|---|---|---|
| 10 | 334.5 | 2387 | 797 | 7.83 | 1.09 | 66.60 | 717.66 |
| 20 | 288 | 3181 | 1773 | 7.74 | 2.58 | 44.25 | 88.67 |
| 50 | 295.5 | 4317 | 2105 | 10.36 | 2.58 | 51.25 | 63.14 |
| 100 | 285 | 5293 | 3759 | 10.04 | 5.09 | 28.99 | 37.59 |
| 500 | 283.5 | 10,337 | 4724 | 14.31 | 5.09 | 54.30 | 42.51 |
| 1000 | 289.5 | 11,856 | 5854 | 15.13 | 6.25 | 50.63 | 5.27 |
| 10,000 | 290.25 | 18,481 | 9344 | 22.23 | 8.1 | 49.44 | 0.84 |
| Sum | - | - | - | - | - | - | 956 |

**Table 6.** Optimized results from simulated optimization model OP1 (genes, 100; population, 80; *n* = 2).

| Return Period | Time (hr) | Peak Inflow (m$^3$/s) | Peak Outflow (m$^3$/s) | Max Water Level (m) | Max Gate Opening (m) | Flood Reduction (%) | Objective Function Value (m$^3$/s) |
|---|---|---|---|---|---|---|---|
| 10 | 288 | 2387 | 1423 | 5.75 | 2.48 | 40.39 | 854.65 |
| 20 | 291 | 3181 | 1645 | 7.25 | 2.48 | 48.28 | 232.87 |
| 50 | 288 | 4317 | 2784 | 8.93 | 3.9 | 35.51 | 305.24 |
| 100 | 290.25 | 5293 | 3126 | 10.73 | 3.9 | 40.94 | 197.89 |
| 500 | 277.5 | 10,337 | 6301 | 13.32 | 7.58 | 39.05 | 378.35 |
| 1000 | 286.5 | 11,856 | 6721 | 14.6 | 7.58 | 43.31 | 127.63 |
| 10,000 | 285 | 18,481 | 12,187 | 20.75 | 11.73 | 34.06 | 73.18 |
| Sum | - | - | - | - | - | - | 2170 |

**Table 7.** Optimized results from simulated optimization model OP2 (genes, 100; population, 80; *n* = 2).

| Return Period | Time (hr) | Peak Inflow (m$^3$/s) | Peak Outflow (m$^3$/s) | Max Water Level (m) | Max Gate Opening (m) | Flood Reduction (%) | Objective Function Value (m$^3$/s) |
|---|---|---|---|---|---|---|---|
| 10 | 382.5 | 2387 | 676 | 8.85 | 0.86 | 71.67 | 406.21 |
| 20 | 280.5 | 3181 | 2223 | 7.42 | 3.45 | 30.11 | 314.69 |
| 50 | 291 | 4317 | 2581 | 9.37 | 3.45 | 40.23 | 282.92 |
| 100 | 289.5 | 5293 | 3268 | 10.52 | 4.16 | 38.27 | 206.86 |
| 500 | 286.5 | 10,337 | 4110 | 15.39 | 4.16 | 60.24 | 246.82 |
| 1000 | 293.25 | 11,856 | 4395 | 17.29 | 4.16 | 62.93 | 83.46 |
| 10,000 | 290.25 | 18,481 | 9105 | 22.85 | 7.72 | 50.74 | 54.67 |
| Sum | - | - | - | - | - | - | 1596 |

**Table 8.** Optimized results from simulated optimization model OP1 (genes, 100; population, 80; $n$ = 3).

| Return Period | Time (hr) | Peak Inflow (m$^3$/s) | Peak Outflow (m$^3$/s) | Max Water Level (m) | Max Gate Opening (m) | Flood Reduction (%) | Objective Function Value (m$^3$/s) |
|---|---|---|---|---|---|---|---|
| 10 | 294 | 2387 | 1206 | 5.97 | 2 | 49.47 | 527.31 |
| 20 | 297 | 3181 | 1401 | 7.69 | 2 | 55.95 | 233.62 |
| 50 | 292.5 | 4317 | 2375 | 10.05 | 3 | 44.98 | 334.02 |
| 100 | 283.5 | 5293 | 3875 | 11.17 | 4.85 | 26.79 | 377.91 |
| 500 | 280.5 | 10,337 | 5635 | 14.26 | 6.25 | 45.48 | 530.53 |
| 1000 | 288.75 | 11,856 | 5959 | 15.56 | 6.25 | 49.73 | 260.40 |
| 10,000 | 288 | 18,481 | 11,033 | 21.03 | 10.25 | 40.30 | 223.78 |
| Sum | - | - | - | - | - | - | 2488 |

**Table 9.** Optimized results from simulated optimization model OP2 (genes, 100; population, 80; $n$ = 3).

| Return Period | Time (hr) | Peak Inflow (m$^3$/s) | Peak Outflow (m$^3$/s) | Max Water Level (m) | Max Gate Opening (m) | Flood Reduction (%) | Objective Function Value (m$^3$/s) |
|---|---|---|---|---|---|---|---|
| 10 | 382.5 | 2387 | 598 | 9.04 | 0.75 | 74.47 | 261.53 |
| 20 | 294 | 3181 | 1538 | 8.7 | 2.05 | 51.50 | 256.53 |
| 50 | 300 | 4317 | 1798 | 11.5 | 2.05 | 58.50 | 252.86 |
| 100 | 292.5 | 5293 | 2673 | 12.32 | 3 | 49.99 | 260.66 |
| 500 | 285 | 10,337 | 4299 | 16.08 | 4.25 | 58.41 | 404.83 |
| 1000 | 292.5 | 11,856 | 4528 | 17.6 | 4.25 | 61.80 | 197.91 |
| 10,000 | 292.5 | 18,481 | 7843 | 24.59 | 6.25 | 57.56 | 159.08 |
| Sum | - | - | - | - | - | - | 1802 |

More specifically, let us assume that the 10-year flood occurs with no prior knowledge on its hydrograph time variation and peak. In another word, the operator observes the water surface elevation and decides on the gate opening based on the rules defined in Tables 3–9 for the simple ten-stage simulation (STSS), OP1, and OP2 strategies. The real-time release from the reservoir, reservoir water surface elevation, and values of gate opening are depicted in Figure 4. As illustrated, the maximum outflow for the STSS, OP1, and OP2 strategies are approximated as 1640 m$^3$/s, 1206 m$^3$/s, and 598 m$^3$/s, respectively. These outflows result in a 6.1 m, 5.97 m, and 9.04 m increase in reservoir water level, respectively. For a 50-year flood, the STSS strategy modifies the gate opening in three stages with a maximum opening of 3.98 m and a peak outflow of 2944 m$^3$/s. Both the OP1 and OP2 strategies, however, modify the gate opening in only two stages with maximum outflow (gate opening) of 2375 m$^3$/s (3 m) and 1798 m$^3$/s (2.05 m), respectively (Figure 5).

In the OP2 strategy, a much better degree of subsidence is obtained for floods with different return periods, compared to the OP1 and the ten-stage model strategies. However, when using the instructions obtained from the OP1 strategy (Tables 8 and 9), the total values of the objective function for floods with a return period of 10 to 10,000 years do not exceed 2488 m$^3$/s. In this case, the percentage of reduction of flood peak increase from 26.79% to 55.95%, which confirms the more effective behavior by using the OP1 strategy. In addition, when using the instructions obtained from the OP2 strategy (Tables 8 and 9), the total values of the objective function for floods with a return period of 10 to 10,000 years do not exceed 1802 m$^3$/s. In this case, the percentage of reduction of the flood peak goes from 49.99% to 74.47%, which also confirms a more effective behavior when using the OP2 strategy. In other words, the results obtained by using the OP2 strategy are much better than those from the ten-stage and OP1 strategies.

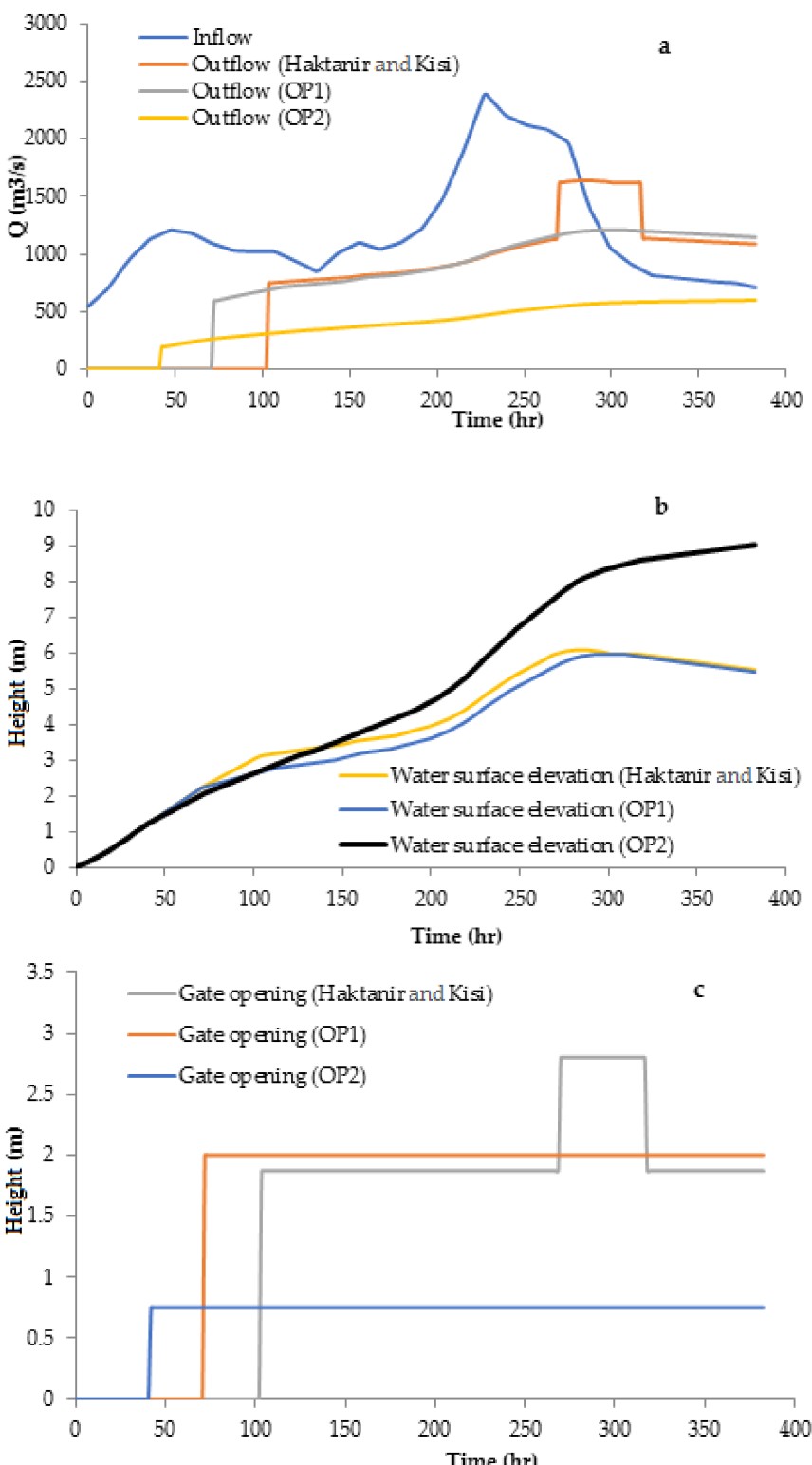

**Figure 4.** Application of both approaches—flood hydrograph (10-year): (**a**) inflow and outflow hydrographs; (**b**) water surface elevation; (**c**) gate opening (*n* = 3).

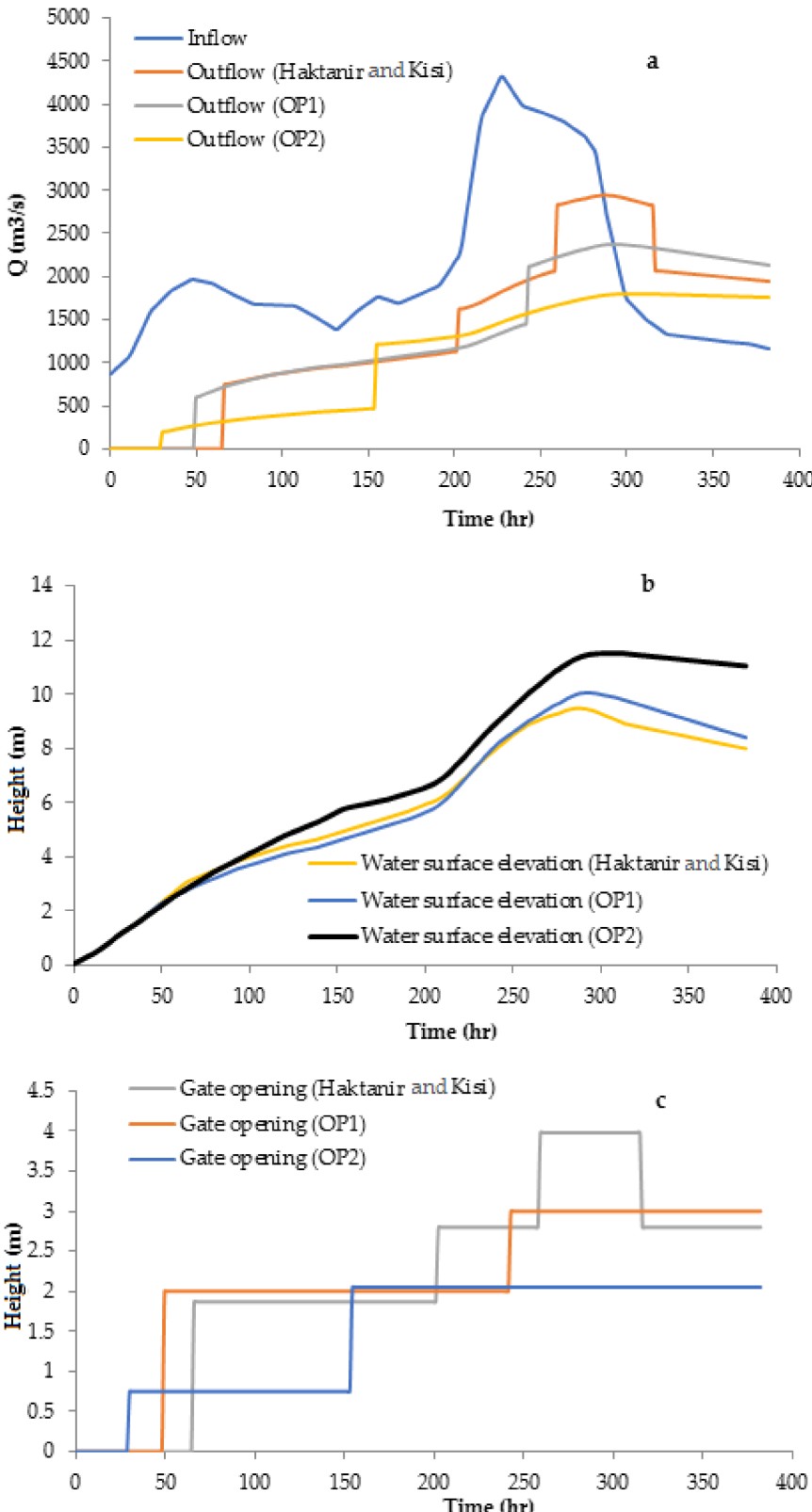

**Figure 5.** Application of both approaches—flood hydrograph (50-year): (**a**) inflow and outflow hydrographs; (**b**) water surface elevation; (**c**) gate opening (*n* = 3).

## 5. Conclusions

This research was launched to simulate an operational model of the spillway gates by two different approaches to enhance flood control using an improved gate operation

strategy. Two ten-stage models were developed and tested for enhancing flood control by a modified gate operation strategy. The control critical levels and corresponding gate openings were optimally determined employing a simulation optimization procedure. The flood management by gate opening was considered as a multiobjective optimization problem in which the operator may wish to minimize the flood damage for all floods with different probabilities of occurrence. In the absence of any economical flood loss measures, an exponential relation was defined to reflect the operator's privileges in floods with different probabilities of occurrence. The performance of the modified OP1 and OP2 models were assessed in addition to Haktanir and Kisi's [18] model to operate the spillway gates of the Karkheh dam. Results showed that the values of critical levels and gate openings calculated by the proposed OP1 and OP2 models outperformed the existing simple ten-stage model in reducing the maximum outflow for various return periods. It was shown that the proposed modeling approach is quite simple and robust in developing more efficient operating rules for gate opening, resulting in more beneficial flood control in gated spillways. It may easily be revised as new information on flood hydrographs becomes available. The proposed operating strategy can be implemented when no flood forecasting data are available, which may be the case in many ungauged rivers. The results of the proposed strategy may be improved if the economic data on flood losses become available. To improve the proposed models and be more practical, model nonlinearities as well as the parameters' uncertainty should be included in the problem.

**Author Contributions:** Conceptualization, F.S., M.N., M.M.N. and M.M.H.; methodology, F.S., M.N., M.M.N. and M.M.H.; software, F.S.; validation, F.S. and M.N.; formal analysis, F.S., M.N., M.M.N. and M.M.H.; investigation, F.S., M.N., M.M.N. and M.M.H.; resources, F.S., M.N., M.M.N. and M.M.H.; data curation, F.S. and M.N.; writing—original draft preparation, F.S. and M.N.; writing—review and editing, F.S. and M.N.; visualization, F.S.; supervision, M.N., M.M.N. and M.M.H.; project administration, F.S., M.N., M.M.N. and M.M.H.; funding acquisition, none. All authors have read and agreed to the published version of the manuscript.

**Funding:** This research received no external funding.

**Institutional Review Board Statement:** Not applicable.

**Informed Consent Statement:** Not applicable.

**Data Availability Statement:** In this paper, the data set belongs to the Karkheh dam located in the south-west of Iran, which is a multi-purpose embankment dam with a 127 m height and a total storage volume of approximately 5900 million $m^3$. Its main objective is to irrigate 320,000 hectares of irrigable land. Hydropower production and flood control have been referred to as its second and third objectives. The spillway is equipped with six 18 m $\times$ 15 m gates installed on a chute spillway. The spillway crest is located at 209 m.a.s.l. and the maximum flood level of the reservoir has been fixed at 234 m.a.s.l. The spillway is designed to discharge a 10,000-year flood with a 1.2 m freeboard. Estimated flood hydrographs with different return periods are presented in Figure 2. The Karkheh dam shares data only with some research groups in some universities in Iran and are confidential. More detailed explanations regarding this data set are given in the section "Model implementation and results".

**Conflicts of Interest:** The authors declare no conflict of interest.

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
