# Peer review of "Multistage Models for Flood Control by Gated Spillway: Application to Karkheh Dam"

_water, doi:10.3390/w14050709_

Round 1
Reviewer 1 Report
General comments
- The topic is very interesting: How to operate gated spillways without information from the catchment or the inflow? This is relevant and very, very difficult. This approach by testing the response in the reservoir and have algorithms and procedures from weighting of possible downstream consequences is very promising.
- The research has a very clear structure from motivation, methodology development, modelling, application and finally evaluation. This is a strong card for the paper.
- The paper is well written and only minor language improvements must be done.
- Of course, the model should have been verified by a case study, but as for most reservoirs field data from large floods are not available. Obviously because these large floods rarely occur…….
- A strict review of the number of digits in all numbers must be done. The paper has many places a ridiculous number of digits. Up to five digits when the main input parameter has one! This indicate a total lack of understanding of the numbers and the accuracy and must be avoided. Mainly from page 9 – 15. Flood values of 2725,84 for estimated floods is just funny. Must be revised all over.
Detailed comments (line by line)
70 …an optimal control problem for….. What is this? And what is it optimal for? Is to avoid costs, avoid downstream consequences, upstream consequences? What is it optimum for?
105 Water laws? Replace the fraise with physical laws or similar, to avoid misunderstanding with domestic/regional/local laws for use of water.
136 ….that do not need flood forecasting….. This is not the right way of expressing it, you will always use furcating data if you have them. Try something like this …operation policies which can be used when flood forecasting is not available……
156 Mainly or only, please clarify? It is only if you assume the upstream parameters but if you assume the spillway capacity it’s not only anymore. Could you write ……only on reservoir water level and spillway behavior/capacity….? Mainly is a bit mysterious when you don’t present the other parameters.
193 and 199 use 0,1 or 10% consequently.
205 In the equations starting from (1) there is a total lack of units. It is very difficult to follow the development of the equations when you don’t see the units. Like in line 205 the j must be in [m] if it is a level (I can see it isn’t), Scr must be in [m3], fi(Qi) must be in [IRR/$/€/or other currency], ED in currency, etc, etc.. Please include units.
286 Equation (19) can be simplified by including 2/3 and 21/2 in the empirical coefficient 0,552. When you already have an empirical coefficient in the equation, you don’t need to keep the accurate numbers separately.
305 a4? Possibly a typing miss or is it a meaning?
339 Just an editorial comment, the …..six 18x15 meter gates…. may lead to some misunderstanding with the x in six so close to the other x’. Maybe not important, but I had to read it twice. If you write ….six similar 18x15 m gates……
359 Here the misleading number of digits appear. Please revise, seriously. An increase of 47.33% on estimated values are very from what you can rely on.
362 Here is also the confusion about the units? You give values, but what is it? Objective function, what is that? Just a number.
Table2 Here is some confusion about the numbers and the units? Gate opening [m] is given as D1, D2…., seems like steps, not meter? OP1-GA and OP2-GA are given without units, but seems to be m.a.s.l.? To the right in the table the OP1-GA and OP2-GA are given with totally different numbers, starting from zero. Please clarify.
Table2 Difficult to understand when you don’t know what Objective functional value is? And far too many digits in the value.
386 It is maybe not necessary to show alle the tables with numbers, I don’t think anyone can extract the full understanding from all the numbers. Please select what is the most import numbers/tables and highlight it.
409 Revise numbers, and please be consequent with the number of digits.
Author Response
Authors’ Response to the Review Comments
Journal: Water
Manuscript #: Water-1576473
Title of Paper: Multi-Stage Models for Flood Control by Gated Spillway: Application to Karkheh Dam
Dear Reviewer
We thank you for your thorough reading of the manuscript and insightful comments. We have provided detailed responses below. The corresponding changes in the manuscript are highlighted in Blue colour.
Reviewer 1:
The topic is very interesting: How to operate gated spillways without information from the catchment or the inflow? This is relevant and very, very difficult. This approach by testing the response in the reservoir and have algorithms and procedures from weighting of possible downstream consequences is very promising.
The research has a very clear structure from motivation, methodology development, modelling, application and finally evaluation. This is a strong card for the paper.
The paper is well written and only minor language improvements must be done.
Of course, the model should have been verified by a case study, but as for most reservoirs field data from large floods are not available. Obviously because these large floods rarely occur…….
We would like to thank the reviewer for the careful and thorough reading of this manuscript and for the thoughtful comments, which has helped us to improve the quality of this manuscript. Our answers to the detailed comments are provided below.
Comment 1:
A strict review of the number of digits in all numbers must be done. The paper has many places a ridiculous number of digits. Up to five digits when the main input parameter has one! This indicate a total lack of understanding of the numbers and the accuracy and must be avoided. Mainly from page 9 – 15. Flood values of 2725,84 for estimated floods is just funny. Must be revised all over.
Response: Agreed. We have revised the number of digits all over the paper including pages 9-15.
Detailed comments (line by line)
Comment 2:
70 …an optimal control problem for….. What is this? And what is it optimal for? Is to avoid costs, avoid downstream consequences, upstream consequences? What is it optimum for?
Response: Thank you for pointing this out. Generally, one may consider the flood management by gate opening as a multi-objective optimization problem in which the operator wishes to minimize the flood damage for all floods with different probability of occurrence. As you mentioned, the aim of optimal control problem is to avoid downstream consequences (the more reduce flood peak outflows the more reduce downstream consequences). The goal is to reduce peak outflows that may lead to downstream flood damage.
Comment 3:
105 Water laws? Replace the fraise with physical laws or similar, to avoid misunderstanding with domestic/regional/local laws for use of water.
Response: We have revised this issue in the revised manuscript.
Comment 4:
136 ….that do not need flood forecasting….. This is not the right way of expressing it, you will always use furcating data if you have them. Try something like this …operation policies which can be used when flood forecasting is not available……
Response: Thank you for pointing this out. We have corrected that in the revised manuscript.
Comment 5:
156 Mainly or only, please clarify? It is only if you assume the upstream parameters but if you assume the spillway capacity it’s not only anymore. Could you write ……only on reservoir water level and spillway behavior/capacity….? Mainly is a bit mysterious when you don’t present the other parameters.
Response: Thanks for your suggestion. We meant the operation policies that only rely on the reservoir water level and spillway capacity/behavior. We have clarified that in the revised manuscript.
Comment 6:
193 and 199 use 0,1 or 10% consequently.
Response: That is right. Done.
Comment 7:
205 In the equations starting from (1) there is a total lack of units. It is very difficult to follow the development of the equations when you don’t see the units. Like in line 205 the j must be in [m] if it is a level (I can see it isn’t), Scr must be in [m3], fi(Qi) must be in [IRR/$/€/or other currency], ED in currency, etc, etc.. Please include units.
Response: We apologize for that. We have corrected all the mentioned issues in the revised manuscript.
Comment 8:
286 Equation (19) can be simplified by including 2/3 and 21/2 in the empirical coefficient 0,552. When you already have an empirical coefficient in the equation, you don’t need to keep the accurate numbers separately.
Response: In the revised manuscript, we have combined them in a single coefficient which is equivalent to 1.63.
Comment 9:
305 a4? Possibly a typing miss or is it a meaning?
Response: Thanks for noting this. It was supposed to be Equation (4-a) that has been corrected in the revised version.
Comment 10:
339 Just an editorial comment, the …..six 18x15 meter gates…. may lead to some misunderstanding with the x in six so close to the other x’. Maybe not important, but I had to read it twice. If you write ….six similar 18x15 m gates……
Response: We have revised it accordingly to avoid confusion for the readers.
Comment 11:
359 Here the misleading number of digits appear. Please revise, seriously. An increase of 47.33% on estimated values are very from what you can rely on.
Response: We are not sure if we clearly understand your concern here, but we are saying that the flood peak, which was , reduces to 9734 . This reduction is equivalent to . Also, the flood peak values for all the return periods are shown in Figure 3.
Comment 12:
362 Here is also the confusion about the units? You give values, but what is it? Objective function, what is that? Just a number.
Response: The unit for the object function is (it is a weighted function of peak outflows) which has been noted in different parts of the revised manuscript.
Comment 13:
Table2 Here is some confusion about the numbers and the units? Gate opening [m] is given as D1, D2…., seems like steps, not meter? OP1-GA and OP2-GA are given without units, but seems to be m.a.s.l.? To the right in the table the OP1-GA and OP2-GA are given with totally different numbers, starting from zero. Please clarify.
Response: This table includes two parts where the left-hand side shows the critical level values for the three operating strategies with the unit m.a.s.l. On the right-hand side of the table, the values of (D1 to D10 are the steps for gate openings) are given again for the three operating strategies. The associated units have been added as well.
Comment 14:
Table2 Difficult to understand when you don’t know what Objective functional value is? And far too many digits in the value.
Response:
Table 2 gives the critical values and also for all the operating strategies and it does not give any information about the object function. However, if you mean Table 3, the objective function unit, which is , has been added in the revised manuscript. Also, number of the digit have been revised to be fairly small and consistent with other parts of the manuscript.
Comment 15:
386 It is maybe not necessary to show all the tables with numbers, I don’t think anyone can extract the full understanding from all the numbers. Please select what is the most import numbers/tables and highlight it.
Response: In the revised tables, we have highlighted the most important columns.
Comment 16:
409 Revise numbers, and please be consequent with the number of digits.
Response: Thank you for this important comment. We have considered this in the revised manuscripts.

Reviewer 2 Report
Reviewed paper deals with the multi-stage models for flood control by gated spillway: The scope of the paper well fits the scope of Water. The paper contains new interesting results important for application. I recommend this paper for publication in Water after minor revision taking into account the following comments:
- There are closely related papers:
- Buber A., Bolgov M. Multi-Criteria Analysis of the “Lake Baikal—Irkutsk
Reservoir” Operating Modes in a Changing Climate: Reliability, Resilience, Vulnerability. Water2021, 13(20), 2879; - Bolgov, M.V., Buber, A.L., Komarovskii, A.A., Lotov, A.V. (2018). Searching for Compromise Solution in the Planning and Managing of Releases into the Lower Pool of the Volgograd Hydropower System. 1. Strategic Planning. Water Resources, 45(5), 819 -826.
Author Response
Authors’ Response to the Review Comments
Journal: Water
Manuscript #: Water-1576473
Title of Paper: Multi-Stage Models for Flood Control by Gated Spillway: Application to Karkheh Dam
Dear Reviewer
We thank you for your thorough reading of the manuscript and insightful comments. We have provided detailed responses below. The corresponding changes in the manuscript are highlighted in Blue colour.
Reviewer 2:
Reviewed paper deals with the multi-stage models for flood control by gated spillway: The scope of the paper well fits the scope of Water. The paper contains new interesting results important for application. I recommend this paper for publication in Water after minor revision taking into account the following comments:
There are closely related papers:
Buber A., Bolgov M. Multi-Criteria Analysis of the “Lake Baikal—Irkutsk
Reservoir” Operating Modes in a Changing Climate: Reliability, Resilience, Vulnerability. Water2021, 13(20), 2879;
Bolgov, M.V., Buber, A.L., Komarovskii, A.A., Lotov, A.V. (2018). Searching for Compromise Solution in the Planning and Managing of Releases into the Lower Pool of the Volgograd Hydropower System. 1. Strategic Planning. Water Resources, 45(5), 819 -826.
Response:
We would like to express our appreciation to the reviewer for reading our paper and the suggestions. We have addressed the above suggested papers in the revised manuscript.

Reviewer 3 Report
The authors in the paper entitled, “Multi-Stage Models for Flood Control by Gated Spillway: Application to Karkheh Dam” discussed simulation of an operational model of the spillway gates by two different approaches to enhance flood control using an improved gate operation. The first strategy “OP1” is based on the reservoir water level as input data The Second strategy “OP2” is based on the observed reservoir water level and flood peak in the nearest upstream gaging station. The authors claimed that in both the approaches no flood forecasting data is required thus they can be adoptable to any flood with any return period. A number of researchers including “Haktanir” have carried out similar studies with different stage models. Therefore, authors need to mention more clearly what is the novelty and limitations of the study. Following are my specific comments which need to be clarified for the acceptance of the paper:
- Kharkheh dam is a multi-purpose dam water level in the reservoir fluctuates with the requirements of power generation and irrigation in different months. Reservoir levels will also fluctuate with the rainfall and other factors in the catchment, then how operator will decide the opening of the gates based on the hydrographs with different return periods.
- Authors have proposed a simulation-optimization procedure for gate openings based on the water levels in reservoir for the management of downstream flood to minimize the flood damage for all floods with different probability of occurrence. How it is possible? It will be trial and error method. Authors have to mention that this is the only dam in the river basin or other dams are also constructed in the upstream which may control the reservoir water level of Kharkheh dam.
- In the second approach Op2, authors have mentioned that in this strategy benefits from both observed reservoir water level and flood peak in the nearest upstream gaging station will be utilized. How, on the basis of one gaging station flood peak will be decided. “Nearest upstream gaging station” is vague statement. Be specific and provide location with justification.
- Methodology needs more information. Authors have provided a general flow chart showing a Genetic Algorithm (GA)-based simulation-optimization model of gated A separate paragraph should be added describing GA and its utility in such type of study. Methodology should be more clear so reader can understand modelling process proposed in this study.
- Models’ validation methods for the estimation of floods should be mentioned.
Author Response
Authors’ Response to the Review Comments
Journal: Water
Manuscript #: Water-1576473
Title of Paper: Multi-Stage Models for Flood Control by Gated Spillway: Application to Karkheh Dam
Dear Reviewer
We thank you for your thorough reading of the manuscript and insightful comments. We have provided detailed responses below. The corresponding changes in the manuscript are highlighted in Blue colour.
Reviewer 3:
The authors in the paper entitled, “Multi-Stage Models for Flood Control by Gated Spillway: Application to Karkheh Dam” discussed simulation of an operational model of the spillway gates by two different approaches to enhance flood control using an improved gate operation. The first strategy “OP1” is based on the reservoir water level as input data. The Second strategy “OP2” is based on the observed reservoir water level and flood peak in the nearest upstream gaging station. The authors claimed that in both the approaches no flood forecasting data is required thus they can be adoptable to any flood with any return period. A number of researchers including “Haktanir” have carried out similar studies with different stage models. Therefore, authors need to mention more clearly what is the novelty and limitations of the study. Following are my specific comments which need to be clarified for the acceptance of the paper:
Response:
In this paper, our proposed methods, i.e. OP1 and OP2 are improved versions of the model by Haktanir. Compared to the model by Haktanir, OP1 model strategy benefits from observed reservoir water level and also uses the well-known Genetic algorithm to obtain the optimal model parameters. OP2 benefits from both observed reservoir water level and flood peak in the nearest upstream gaging station. Using Genetic algorithm, the critical levels and the optimal gate opening of Karkheh dam gated spillways have been optimized. Also, none of the proposed policies needs flood forecasting data, thus, making them adaptable to any flood with any return period. Results have shown that values of critical levels and gate openings calculated by the proposed OP1 and OP2 models outperform the already existing simple ten-stage model in reducing the maximum outflow for different return periods. Regarding the limitations, the models can be improved by considering model nonlinearities as well as model parameters uncertainties to make the models more robust. We have added more to the Introduction and Conclusion sections to make the novelties clearer for the readers.
Comment 1: Kharkheh dam is a multi-purpose dam water level in the reservoir fluctuates with the requirements of power generation and irrigation in different months. Reservoir levels will also fluctuate with the rainfall and other factors in the catchment, then how operator will decide the opening of the gates based on the hydrographs with different return periods.
Authors have proposed a simulation-optimization procedure for gate openings based on the water levels in reservoir for the management of downstream flood to minimize the flood damage for all floods with different probability of occurrence. How it is possible? It will be trial and error method.
Response:
Part A: Yes, to decide on the gate opening, the operator only need to consider the reservoir water level where often there is no considerable fluctuations there. The operator looks at the reservoir water level and compare it with the critical levels and then decide how much the gates should be opened.
Part B: The minimization problem is solved using the idea of genetic algorithm which is an iterative probabilistic search method. Please kindly note, an implementation of a genetic algorithm begins with a population of (typically random) chromosomes. One then evaluates these structures and allocates reproductive opportunities in such a way that those chromosomes which represent a better solution to the target problem are given more chances to 'reproduce' than those chromosomes which are poorer solutions. The goodness of a solution is typically defined with respect to the current population.
We have added a paragraph about the genetic algorithm in the revised version.
Comment 2: Authors have to mention that this is the only dam in the river basin or other dams are also constructed in the upstream which may control the reservoir water level of Kharkheh dam.
Response: To the best of authors knowledge, Karkheh dam is the only dam in the river basin and we have added this note to the results section of the revised manuscript.
Comment 3: In the second approach Op2, authors have mentioned that in this strategy benefits from both observed reservoir water level and flood peak in the nearest upstream gaging station will be utilized. How, on the basis of one gaging station flood peak will be decided. “Nearest upstream gaging station” is vague statement. Be specific and provide location with justification.
Methodology needs more information. Authors have provided a general flow chart showing a Genetic Algorithm (GA)-based simulation-optimization model of gated A separate paragraph should be added describing GA and its utility in such type of study.
Response:
Part A: In this method, a river flow measuring station is required at the connection (or close to) of the river to the reservoir. Also, we have assumed that a system for transferring measured data to the control site is available. In this way, the information about the flood peak would be available. We have clarified this in the revised version at page 4, before starting the Methodology section.
Part B: Thanks for this important note. We have added a separate paragraph to page 8 about the Genetic algorithm to explain the algorithm and its utility for this study.
Comment 4: Methodology should be more clear so reader can understand modelling process proposed in this study.
Response: To make the proposed method clearer for the readers, please kindly note that we have added more details to different parts of the Methodology in the revised manuscript including genetic algorithm.
Comment 5: Models’ validation methods for the estimation of floods should be mentioned.
Response: Sorry the question is not very clear to us, but to evaluate the models’ performances we have considered that the objective is to minimize the peak flow. In the results section for all the models we have compared their flood reduction% by all the models for different return periods. Also, in Table 2, we have brough the obtained gate opening steps (D1-D10) and critical levels. We can see the superiority of OP2 and also superiority of both OP1 and OP2 compared to the model proposed by Hektanir. We have highlighted the validation results in Table 3 -Table 9.
Round 2
Reviewer 3 Report
Authors have improved manuscript as per suggestions. it can be accepted for the publication.
This manuscript is a resubmission of an earlier submission. The following is a list of the peer review reports and author responses from that submission.